# Food insecurity in a Brazilian transgender sample during the COVID-19 pandemic

**Sávio Marcelino Gomes** [1]*, **Michelle Cristine Medeiros Jacob**[2], **Viviany Moura Chaves**[2], **Luciana Maria Pereira de Sousa**[3], **Marcos Claudio Signorelli**[4], **Daniel Canavese de Oliveira**[5], **Clélia de Oliveira Lyra**[3], **Luiz Roberto Augusto Noro**[3]

1 Department of Nutrition, Federal University of Paraíba, João Pessoa, Paraíba, Brazil, 2 Department of Nutrition, Laboratório Horta Comunitária Nutrir, Universidade Federal do Rio Grande do Norte, Natal, Rio Grande do Norte, Brazil, 3 Programa de Pós-Graduação em Saúde Coletiva, Universidade Federal do Rio Grande do Norte, Natal, Rio Grande do Norte, Brazil, 4 Department of Public Health, Universidade Federal do Paraná, Curitiba, Paraná, Brazil, 5 Department of Public Health, Universidade federal do Rio Grande do Sul, Porto Alegre, Rio Grande do Sul, Brazil

* savio.gomes@academico.ufpb.br

## Abstract

Transgender people often live with social vulnerability, largely promoted by gender-based prejudice. Our aim in this article was to raise preliminary data on how the COVID-19 pandemic and perceived prejudice have contributed to the problem of food and food insecurity in the transgender communities in Brazil. We conducted a web-based cross-sectional study, in which 109 transgender people from all regions of Brazil participated. We used the Chi-Square test and Poisson regression modeling with robust variance to estimate the association between food insecurity and the investigated factors. In our sample, 68.8% of transgender people experienced food insecurity, of these, 20.2% experienced severe food insecurity. Our results showed that the difficulties in purchasing food in the transgender community predate the COVID-19 pandemic, yet that the restrictive measures adopted have also impacted overall access to quality food. However, the main explanations for food insecurity were income and employment. In predicting food insecurity, the experiences of prejudice must be considered, and give rise to the hypothesis that specific conditions to which transgender people are exposed explain, to some degree, their vulnerability to food insecurity.

## Introduction

In countries with fragile social support policies, the establishment of social distancing measures, needed to contain the COVID-19 pandemic, has only widened existing disparities. In the United States, since the start of the pandemic, inadequate state intervention to support vulnerable families has already resulted in increased food insecurity [1]. According to the Food and Agriculture Organization (FAO), a family is experiencing food insecurity when they do not have permanent access to culturally and nutritionally adequate food [2]. Brazil as well, with its limited support policies for socially vulnerable families, has also seen an increase in the prevalence of food insecurity. Currently, more than half of the Brazilian population experienced some degree of food insecurity [3].

**Funding:** This work was supported by Coordenação de Aperfeiçoamento de Pessoal de Nível Superior (CAPES), for the Ph.D. scholarship to SMG (grant numbers 88887.505839/2020-00), by the CNPq through a research grant to MCMJ (402334/2021-3) and a research productivity scholarship also awarded to MCMJ (306755/2021-1). The funders had no role in study design, data collection and analysis, decision to publish, or preparation of the manuscript.

**Competing interests:** The authors have declared that no competing interests exist.

Women are four times more likely than men to experience food insecurity worldwide. This inequality can be attributed to two main factors: 1) food insecurity often results from negative social, political, and economic conditions that disproportionately affect certain groups, and 2) women in many countries are disadvantaged compared to men in terms of education, income, and support networks, among other things [4]. However, gender-based studies on food insecurity have often adopted a categorical paradigm, which assumes the existence of only two gender categories: men and women. This categorical thinking has informed statistical differences in studies and health policies, such as the higher rates of hunger among women compared to men. The main limitation of categorical thinking is that it underestimates diversity within gender categories, such as the differences between hegemonic and subordinate masculinities or between lesbian and heterosexual women. A relational approach to gender provides a more sophisticated means of identifying the magnitude of social problems. This approach considers the effects of large-scale patterns found in politics, economics, and culture on masculinity and femininity patterns among populations, regardless of their biological sex [5–7].

In addition to the categorical thinking prevalent in population-based research on food insecurity, we also believe that considering the division between cisgender and transgender identities can reveal specific exposure factors for minority gender populations. Studies prior to the pandemic suggest that in gender minorities, such as the transgender population, food insecurity is an important indicator of social vulnerability [8–10]. This population has been historically excluded from the health, education, and social assistance sectors, resulting in worse rates of violence, stigma, and long-term illness, as well as school dropout and unemployment [11]. These factors, together with perceived prejudice [12], which occurs when transgender people identify episodes of exclusion against themselves, make the transgender population vulnerable to food insecurity. However, we lack baseline data to monitor the effects of the COVID-19 pandemic in this specific group, or to investigate predictive factors for food insecurity in the transgender community. The main problem associated with the lack of information is the invisibility of this group in terms of measures (i.e. in public policies and contingency plans) that might help reduce the impact of the pandemic and the consequent perpetuation of historically unequal situations. Since the beginning of the pandemic, gender researchers have called for monitoring the health condition of transgender persons, considering their specificities [13].

Thus, our objective was to estimate the frequency of food insecurity in a sample of transgender people and to investigate its associations with dietary change and with prejudice as perceived in Brazil during the COVID-19 pandemic.

## Methods

### Study design

We conducted an online cross-sectional study, using a non-probabilistic sample.

### Participants

We conducted an online recruitment of adult transgender people (> 18 years old) who lived in Brazil at the time of the study. We asked people about their gender identity (trans men, trans women, travestis (a Brazilian identity), gender non-conforming, or other non-cisgender identities. For the analyses, we considered them all as transgender people.

### Data collection

We used social media such as Instagram, Twitter, Facebook, and WhatsApp to recruit participants between the months of October and December 2020. We used social media platforms

because of their potential for recruiting members of stigmatized groups successfully [14]. We gained the support of national LGBT social movement networks, public outpatient clinics specializing in healthcare for the transgender population, and researchers with a network of participants contacted in previous research. We considered as eligible for the study people who met the following criteria: (1) living in the national territory, (2) being 18 years of age or older, and (3) identifying themselves as being a transgender person. Participation in the survey was completely voluntary and confidential.

## Instruments and variables

Our theoretical model (Fig 1) is based on the assumptions detailed below. The COVID-19 pandemic has presented considerable impacts on food and nutritional insecurity of the Brazilian population, and may also have affected the transgender community [3]. The main factor that differentiates the transgender community from the cisgender population is an prejudice based on gender identity [12]. Work, education, and income [3], as well as transgender vulnerabilities [15,16] are important variables in explaining food insecurity.

Our dependent variable was food insecurity. To measure it, we used the Brazilian Food Insecurity Scale (EBIA), in its reduced eight item version [17]. The quality and psychometric properties of this abbreviated scale in the Brazilian context have been evidenced in national surveys since 2004, making the EBIA the official Brazilian tool to determine food insecurity levels in the population. We classified degrees of food insecurity based on the final scale score, with the following cutoff points: Food security (0), mild (1–3), moderate (4–5), and severe food insecurity (6–8). Afterward, we grouped the different levels of food insecurity into a single category. The EBIA assesses an individual's perception of their food insecurity for the three months prior to application.

As independent variables, we considered socioeconomic characteristics (income, education, and work), any dietary changes since the beginning of the COVID-19 pandemic, and perceived prejudice. We asked participants for self-declarations of gender and color. We used tertiles to analyze income. For education, we considered access to university education and basic education (primary and secondary school). We measured dietary changes based on three variables: 1) ability to purchase food; 2) access to diversified food; 3) access to food one considers healthy. These three variables were measured with reference to the period before the COVID-19 pandemic and the period after the onset of the pandemic. We classified these changes and created three variables: "Change in the ability to purchase food," "change in food diversity," and "change in the ability to access healthy food." These were divided into three categories: "no difficulties before or after," "began to have difficulty," and "it remains as difficult as it was

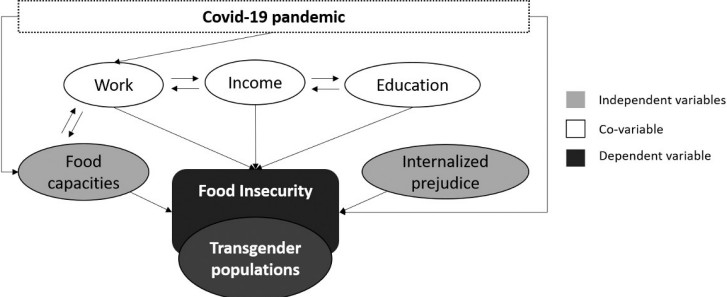

**Fig 1. Theoretical analysis model with the relationships between food insecurity, independent variables, and covariates.**

before." Likewise, we measured employment losses from the variables "worked before the pandemic" and "worked after the onset of the pandemic," and further categorized a new dummy variable we called "lost my job during the COVID-19 pandemic" (Yes or No).

We measured gender-based prejudice using the Perceived Prejudice Scale [18], which measures the frequency of perceived experiences of sexual fetishism, job rejection, verbal violence, physical violence, and police violence (because of being transgender); as measured from 0 = never, 1 = once or twice, 2 = often, and 3 = always. We categorized each total score based on a method suggested by the authors who validated the scale, into quartiles, for low, medium, high, and extreme perceived prejudice.

## Statistical analysis

We described the study variables in absolute and relative frequency. We performed bivariate analysis using Pearson's Chi-square test and standardized residuals to verify association strengths between the variable categories. We used Poisson regression with robust variance to build an association model between food insecurity, dietary changes as a result of the COVID-19 pandemic, and perceived prejudice. This was adjusted for income, education, and the impact of the pandemic on employment. This method had already been suggested for cross-sectional studies [19–22] because of its potential to directly measure the Prevalence Ratio (PR) and to use its respective corrected 95% confidence intervals (95% CI). Further, a high frequency for the food insecurity outcome might overestimate association measures when using concurrent methods (i.e. Odds Ratio). The theoretical model we tested is described in Fig 1. The variables in the model were selected by the Forward Selection method, using epidemiologic and statistical criteria. We evaluate the predictive power using the ROC Curve, considering the model's probability of estimating the food insecurity in the sample.

## Ethics

We followed the Brazilian Resolution No. 466/12 on ethical research conduct, in addition to the recommendations for research with a transgender population [23] and the Declaration of Helsinki as revised in 2013. All written participants' consent was obtained. Our research was approved by the Ethics and Research Committee of Potiguar University (Approval No. 4,312,698).

## Results

### Population characteristics

Our questionnaire was answered by 112 transgender people. After excluding duplicates, 109 participants remained, 53.2% (95% CI 43.69–62.50) trans men, 29.4% (95% CI 21.47–38.72) trans women, 6.4% (95% CI 3.01–13.00) travestis, 7.3% (95% CI 3.67–14.43) non-binary, and 3.7% (95% CI 1.36–9.51) other, different identities. These participants were distributed across all of Brazil: 51.4% (95% CI 41.91–60.75) in the Northeast, 8.3% (95% CI 4.31–15.25) in the North, 5.50% (95% CI 2.41–11.85) in the Midwest, 14.68% (95% CI 9.12–22.78) in the Southeast, and 20.18% (95% CI 13.59–28.91) in the South. Of these, 58.7% (95% CI 49.12–67.69) declared themselves as black (black or brown). The first tertile of the population reported a per capita income of up to 91.58 USD per month, the second tertile reported a per capita income of between 91.59 USD and 240.14 USD per month, and the third tertile reported a monthly per capita income of 240.15 USD and above. As for education, 55% had university education (including incomplete and complete) and 45% had only basic education (primary and secondary).

Reliably characterizing the Brazilian transgender community is still a challenge. However, our aim in this study was only to provide initial data on the phenomenon of food insecurity and its associated factors. We believe that, though preliminary, our estimates can help other researchers to carry out more robust sampling and sampling plans.

## Food insecurity and potential effects of the COVID-19 pandemic

Of the study population, 68.8% (95% CI 59.37–76.9) experienced some degree of food insecurity, of which 20.2% (95% CI 13.59–28.91) experienced severe food insecurity (Fig 2). When evaluating dietary changes caused by the COVID-19 pandemic within the sampling, we saw that the pandemic affected food purchasing power by 15.6% (95% CI 9.85–23.81), food diversity by 15.7% (95% CI 9.85–23.81), and access to healthy foods by 11% (95% CI 6.30–18.54).

## Food insecurity and prejudice based on gender

Self-reported gender bias was quite frequent in our study population. Of the different manifestations, gender-based fetishism was one of the most frequent forms of prejudice in the daily life of the community, with 11% reporting feeling affected by this bias always, and 38.5% reporting feeling affected often. This was followed by the perception of rejection at work due to gender prejudice, reported in our sample with a frequency of 10%, and as often as 26.6%. Gender-based physical violence was the least reported, although participants who experienced food insecurity reported higher frequencies of simple prejudices, whether strong or extreme. The opposite behavior occurred with participants having food security, who mainly reported experiences categorized as low prejudice (Table 1).

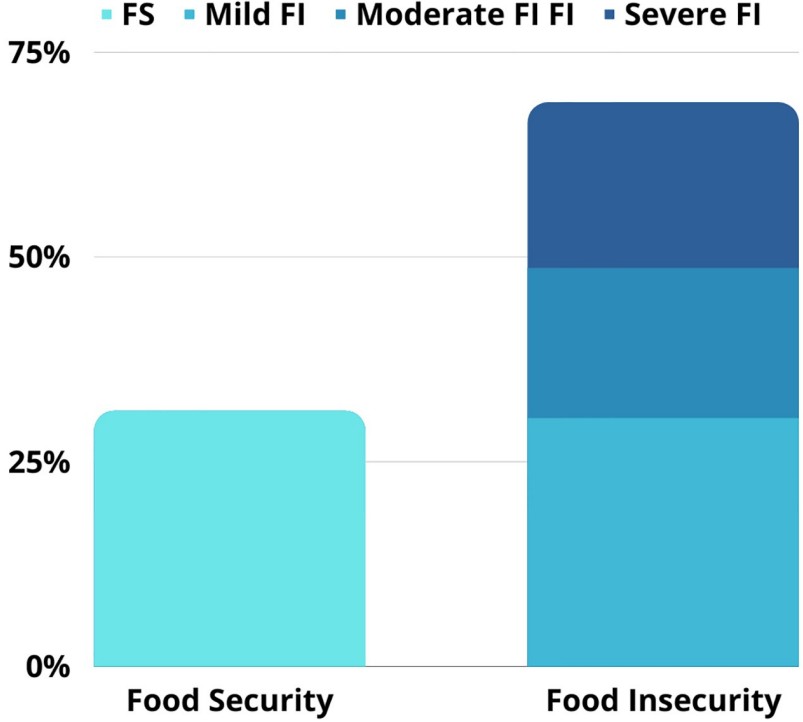

**Fig 2. Relative food access frequencies in the transgender community.**

**Table 1. Association between income, education, employment, prejudice, and changes in diet during the COVID-19 pandemic and food insecurity.**

| Variables | FS | FI | P value |
|---|---|---|---|
| | n (%) | n (%) | |
| **Per capita income** | | | <0.01* |
| 3rd tertile | 21 (61.8) | 15 (20.0) | |
| 2nd tertile | 10 (29.4) | 25 (33.3) | |
| 1st tertile | 3 (8.8) | 35 (46.7) | |
| **Education** | | | <0.01* |
| University education | 25 (73.5) | 35 (43.7) | |
| Basic education | 9 (26.5) | 40 (53.3) | |
| **Employment** | | | <0.01* |
| Yes | 14 (41.2) | 8 (10.7) | |
| No | 20 (58.8) | 67 (89.3) | |
| **Changes in the ability to purchase food during the COVID-19 pandemic** | | | <0.01* |
| No | 32 (94.1) | 51 (68.0) | |
| Yes | 2 (5.9) | 24 (32.0) | |
| **Changes in food diversity during the COVID-19 pandemic** | | | <0.01* |
| No | 31 (91.2) | 43 (57.3) | |
| Yes | 3 (8.8) | 32 (42.7) | |
| **Changes in ability to access healthy food during the COVID-19 pandemic** | | | <0.01* |
| No | 30 (88.2) | 39 (52.0) | |
| Yes | 4 (11.8) | 36 (48.0) | |
| **Perceived prejudice** | | | 0.08 |
| Low | 15 (44.1) | 16 (21.3) | |
| Average | 5 (14.7) | 11 (14.7) | |
| Strong | 8 (23.5) | 23 (30.7) | |
| Extreme | 6 (17.6) | 25 (33.3) | |

FS: Food Security; FI: Food Insecurity;

*p<0,05.

## Food insecurity associated factors

In our bivariate analysis, income, education, work, and factors related to the pandemic were associated with food insecurity (p<0.05), but prejudice was not (Table 1). According to which, the 1st income tertile, access only to basic education, and being unemployed within the COVID-19 period were the socioeconomic categories most associated with food insecurity. We used the crude variables to food changes to fulfill the chi-square test. People who reported difficulties to purchase food and accessing diverse and healthy food during the COVID-19 pandemic were associated with food insecurity. In relation to experiences of prejudice, despite the absence of any association, the group reporting a lower frequency of prejudice presented a higher probability of experiencing food insecurity.

Our theoretical model explains up to 88.46% (CI 82–95) of food insecurity in our sample of Brazilian transgender individuals, in according to ROC area. The per capita income and employment are the best predictors to food insecurity prevalence (Table 2). As for income, its progressive reduction presents a higher PR for food insecurity, an inverse relationship. Being unemployed at the time of the survey also contributed to a greater PR. Regarding the variables on food changes during the COVID-19 pandemic, the model demonstrates that persons who

**Table 2. Prevalence ratios and confidence intervals for associations between changes in feeding capacity (before and during) the COVID-19 pandemic, and for food insecurity, controlled for income and education.**

| Variables | PR Raw | PR Adjusted | CI (95%) | P value |
|---|---|---|---|---|
| **Per capita income** | | | | |
| 3rd tertile | 1 | 1 | | |
| 2nd tertile | 1.71 | 1.71 | 1.15–2.55 | <0.01* |
| 1st tertile | 2.21 | 1.78 | 1.21–2.63 | <0.01* |
| **Education** | | | | |
| University education | 1 | 1 | | |
| Basic education | 1.06 | 1.02 | 0.81–1.28 | 0.88 |
| **Employment** | | | | |
| Yes | 1 | 1 | | |
| No | 2.12 | 1.82 | 1.10–3.02 | 0.02* |
| **Change in the ability to purchase food during the COVID-19 pandemic** | | | | |
| No | 1 | 1 | | |
| Yes | 1.51 | 1.15 | 0.95–1.139 | 0.16 |
| **Change in food diversity during the COVID-19 pandemic** | | | | |
| No | 1 | 1 | | |
| Yes | 1.57 | 1.20 | 0.94–1.52 | 0.14 |
| **Change in ability to access healthy food during the COVID-19 pandemic** | | | | |
| No | 1 | 1 | | |
| Yes | 1.59 | 1.25 | 0.97–1.62 | 0.09 |
| **Perceived prejudice** | | | | |
| Low | 1 | 1 | | |
| Moderate | 1.36 | 1.34 | 0.87–2.06 | 0.19 |
| Strong | 1.81 | 1.29 | 0.93–1.79 | 0.13 |
| Extreme | 1.68 | 1.16 | 0.84–1.61 | 0.37 |

PR: Prevalence Ratio; CI: Confidence Intervals;

*p<0,05.

were difficulties purchasing food, accessing food diversity, and accessing healthy food were respectively 1.15, 1.20, and 1.25 more chances of food insecurity. In the final model, prejudice remained without significance. Our regression model demonstrated the Deviance goodness-of-fit 41.53, with chi2 > 0.05, and the Akaike's information criterion 213.53.

## Discussion

In our study population, factors related to the COVID-19 pandemic further aggravated common vulnerabilities experienced before the pandemic. Prior to the pandemic, difficulties in the ability to purchase food contributed to a higher frequency of food insecurity in our sample. During the pandemic, limited access to healthy and diverse foods increased this frequency. In our final model, the main factors affecting food insecurity were income and employment. The frequency of perceived prejudice was not associated with food insecurity.

Our research fills three gaps in the scientific literature about food insecurity in the Brazilian transgender community: 1) an estimate of the frequency of food insecurity in this population; 2) the impact of the pandemic on access to food; and 3) the association between gender-based prejudice and food insecurity. Our estimated food insecurity frequency of 68.8% in a community sample will allow further research to use sampling and sampling plans that ensure

randomness and population representativeness, which are important for generalizing conclusions. Further study must consider the effects of the COVID-19 pandemic on the transgender diet. However, a lack of baseline data would preclude such a comparison. Finally, we also tested the association between perceived prejudice and food insecurity. The lack of association we found may indicate that the factors related to food insecurity in the Brazilian trans community may be more structural ones, linked to public policies and social protection models, and thus less perceived at the individual level.

In Brazil, women experience higher levels of food insecurity than men, but we do not have national data on transgender people. In a survey conducted during the same period as our research, the prevalence of food insecurity among women was 64.1%, while among men it was 47.5% [3]. Unequal relationships in the economy, education, and support networks contribute to the exacerbation of gender inequalities in determining food insecurity [4]. Our survey showed a higher frequency of food insecurity among transgender people compared to cisgender men and women interviewed, however, these data need to be confirmed in a national population-based survey.

Transgender people are commonly vulnerable to food insecurity. Data from a study carried out in the United States, and using a population and recruitment method similar to ours, revealed a food insecurity prevalence of 10 percentage points above that found in our Brazilian study [8]. When interviewed, transgender people experiencing food insecurity attribute the difficulty in accessing food to limited opportunities for education, training, and employment due to gender stigma. Financial help from family members is generally not an option, due to exclusion from the family as a result of breaking gender norms that are widespread in Western culture. Likewise, this community is also excluded when trying to access local food assistance, especially when there is no state mediation [9].

In our study population, the experience of food insecurity was associated with the level of living and working conditions of social determinants of health. Unemployment and income were positively associated with food insecurity. This relationship occurs directly and is manifested by the difficulty of physical access to food. In our bivariate analysis, education was also associated with food insecurity, however, the association did not hold when we included other social determinants such as employment. Our interpretation is that unemployment overlapped with access to higher education. In our population of transgender individuals, 43.7% of those experiencing food insecurity had access to higher education, but 89% of those experiencing food insecurity were unemployed.

The political and economic scenario in Brazil has also contributed to the worsening of food insecurity in the transgender population and to gender-based prejudice. Since 2004, public sector agencies have pursued policies to promote food security and combat gender discrimination. However, several crises since 2015 have led towards establishment of a neoliberal state and dismantling of those public policies [24,25]. One of the best measured results in the country was the progressive increase in food insecurity after 2018, with food insecurity levels returning to the levels of 2004, before the establishment of the most important food security policies [3].

Transgender communities are historically exposed to experiences of prejudice which mediate the vulnerability of this population to various social problems, such as hunger. According to the Minority Gender Stress Theory [26], research comparing cis and transgender people demonstrates that the trans community experienced higher rates of social rejection by people and institutions. This rejection negatively influences levels of education, income, and social support from friends, family, and the state. This research supports the hypothesis that the determinants of gender inequality in food insecurity, including income, work, education, and social support, may be even more important to the transgender population due to the role of social rejection in worsening them.

In our study, we measured perceived prejudice at the individual level, which does not seem to adequately explain the link between food insecurity and sexual and gender minority experiences of prejudice. Perception of prejudice modulates in relation to individual social conditions, and income and education are important factors in one's perceptions of oppression experienced in everyday life [27]. Further, the relationship between stigma and food insecurity seems not to be located in the individual sphere, but systemically, since in trans communities, economic and political factors interfere collectively in one's quality of life, employment opportunities, education, and income [15,28].

The quality of life of transgender people is associated, in many cases, with gender transition, which has costs that can compete with food costs. About 80% of transgender people require gender affirming procedures, which range from name changes in official documents, to the use of hormones, to surgical procedures [29]. The costs of these procedures vary depending on the level of social protection afforded for transgender people in each country. In Brazil, many of these procedures can be performed free of charge in the national health system, but they are not yet widely available and long processing times of at least 2 years are common [25]. In other countries, payments and co-payments for procedures can reach up to 23,000 USD [30]. Further, even when their needs are covered by public health systems, barriers of access, stigma, and discrimination can prevent transgender people from accessing covered services, necessitating payments to private providers for required services [31,32]. As with all personal costs, direct health costs are associated with food insecurity [33], so the need to perform gender affirming procedures can increase this insecurity [34].

The COVID-19 pandemic significantly affected access to food for the transgender population. Brazilian data reveals that the prevalence of food insecurity in the general population increased from 36.6% in 2018 to 55.2% by the end of 2020 [3]. In our population, the prevalence of severe food insecurity was 20%, double the national estimate of 9%. Research on the impact of the pandemic on the living conditions of the LGBT population reinforces our hypothesis. In a sample of 10,065 people from the sexual and gender minority community, one in five people had a source of income. One out of four transgender people had lost their job due to the pandemic, and, of these, 53% would not have been able to meet basic needs for more than a month. Added to these financial difficulties, the weakening of support networks during the pandemic further intensified the state of vulnerability [16]. In addition, that study documents declining mental health, increasing violence, and deprivation of social protection of transgender people during the pandemic [35].

In our study population, access to food, and the quality and diversity of diets were affected by the COVID-19 pandemic. Studies prior to the pandemic showed that trans communities with difficult access to food began to change their consumption towards ultra-processed, high-calorie, and low-cost foods, this as a way to guarantee quantity, even while reducing quality [36]. Consumption profile changes as a result of physical insecurity or prejudice are common [37], and may explain the dietary changes in our sample.

We highlight as a limitation of our study, the convenience sampling and the online application form. However, we justify their use on account of the lack of previous studies permitting knowledge of the frequency of food insecurity in the Brazilian trans community, and also by concerns related to risk of COVID-19 transmission. In addition, we limited some variables in the regression model (i.e., race and location) because our sample is small, so we needed to select the most important variables, as we explain in our theoretical model. We understand that these questions limited our power of generalization, but we consider the study to be an important first step towards forming a more concrete Brazilian research agenda on the subject.

## Conclusion

Transgender people live in conditions of vulnerability to food insecurity and the indicator is important for monitoring living conditions in this community. Relative to the general population, the frequency of food insecurity was high, and associated (in part) with the impacts of the first year of the COVID-19 pandemic. Food insecurity is a multifactorial problem, and social exclusion is considered one of its causes. Therefore, experiences of prejudice must be considered a contributing factor to economic inequalities and raise the hypothesis that specific conditions to which transgender people are exposed partially explain their food insecurity. To address this, we recommend further studies using methodologies that can demonstrate how prejudice impacts the income and work of the transgender population. Additionally, we suggest conducting studies with larger sample sizes to test the association between different categories of prejudice and food insecurity.

Based on our study, nutritional-epidemiological research can test hypotheses involving determination of food insecurity in the transgender community. Our frequency estimations can be used in sampling plans and sampling forms, allowing study designs that take into account randomness and sample representativeness. Thus, previous experience should support future research designs, to take into account the specificities of research within the transgender population, and important predictors and/or covariates to be considered.

## Acknowledgments

The authors thank Dr. Angelo Roncalli, Dr. Marcos Pereira, and Dr. Rodrigo Viana for his methodological support, and Teacher Jonathan for his language support.

## Author Contributions

**Conceptualization:** Michelle Cristine Medeiros Jacob, Viviany Moura Chaves, Luciana Maria Pereira de Sousa, Marcos Claudio Signorelli.

**Data curation:** Sávio Marcelino Gomes, Michelle Cristine Medeiros Jacob.

**Formal analysis:** Sávio Marcelino Gomes, Michelle Cristine Medeiros Jacob.

**Funding acquisition:** Sávio Marcelino Gomes, Michelle Cristine Medeiros Jacob.

**Investigation:** Sávio Marcelino Gomes, Michelle Cristine Medeiros Jacob.

**Methodology:** Sávio Marcelino Gomes, Michelle Cristine Medeiros Jacob.

**Project administration:** Sávio Marcelino Gomes, Michelle Cristine Medeiros Jacob.

**Resources:** Sávio Marcelino Gomes, Michelle Cristine Medeiros Jacob.

**Software:** Sávio Marcelino Gomes, Michelle Cristine Medeiros Jacob.

**Supervision:** Sávio Marcelino Gomes, Michelle Cristine Medeiros Jacob.

**Validation:** Sávio Marcelino Gomes, Michelle Cristine Medeiros Jacob, Viviany Moura Chaves, Luciana Maria Pereira de Sousa, Marcos Claudio Signorelli, Daniel Canavese de Oliveira, Clélia de Oliveira Lyra, Luiz Roberto Augusto Noro.

**Visualization:** Sávio Marcelino Gomes, Michelle Cristine Medeiros Jacob.

**Writing – original draft:** Sávio Marcelino Gomes, Michelle Cristine Medeiros Jacob, Viviany Moura Chaves, Luciana Maria Pereira de Sousa, Marcos Claudio Signorelli, Daniel Canavese de Oliveira.

**Writing – review & editing:** Sávio Marcelino Gomes, Michelle Cristine Medeiros Jacob, Viviany Moura Chaves, Luciana Maria Pereira de Sousa, Marcos Claudio Signorelli, Daniel Canavese de Oliveira.

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
