## [Decision Letter · Decision Letter 0]

7 Feb 2023

PONE-D-23-00876Food insecurity in a Brazilian transgender sample during the COVID-19 pandemicPLOS ONE

Dear Dr. Gomes,

Thank you for submitting your manuscript to PLOS ONE. After careful consideration, we feel that it has merit but does not fully meet PLOS ONE’s publication criteria as it currently stands. Therefore, we invite you to submit a revised version of the manuscript that addresses the points raised during the review process. Please submit your revised manuscript by Mar 24 2023 11:59PM. If you will need more time than this to complete your revisions, please reply to this message or contact the journal office at plosone@plos.org. Please include the following items when submitting your revised manuscript:A rebuttal letter that responds to each point raised by the academic editor and reviewer(s). You should upload this letter as a separate file labeled 'Response to Reviewers'.A marked-up copy of your manuscript that highlights changes made to the original version. You should upload this as a separate file labeled 'Revised Manuscript with Track Changes'.An unmarked version of your revised paper without tracked changes. You should upload this as a separate file labeled 'Manuscript'.If applicable, we recommend that you deposit your laboratory protocols in protocols.io to enhance the reproducibility of your results. Protocols.io assigns your protocol its own identifier (DOI) so that it can be cited independently in the future. For instructions see: https://journals.plos.org/plosone/s/submission-guidelines#loc-laboratory-protocols. Additionally, PLOS ONE offers an option for publishing peer-reviewed Lab Protocol articles, which describe protocols hosted on protocols.io. Read more information on sharing protocols at https://plos.org/protocols?utm_medium=editorial-email&utm_source=authorletters&utm_campaign=protocols.

We look forward to receiving your revised manuscript.

Kind regards,

Cesar Infante Xibille, Ph.D

Academic Editor

PLOS ONE

“YES - This work was supported by Coordenação de Aperfeiçoamento de Pessoal de Nível Superior (CAPES), for the PhD scholarship (grant numbers 88887.505839/2020-00).”

Reviewers' comments:

Reviewer's Responses to Questions

**Comments to the Author**

1. Is the manuscript technically sound, and do the data support the conclusions?

Reviewer #1: Partly

Reviewer #2: Partly

2. Has the statistical analysis been performed appropriately and rigorously? 

Reviewer #1: Yes

Reviewer #2: Yes

3. Have the authors made all data underlying the findings in their manuscript fully available?

Reviewer #1: Yes

Reviewer #2: No

4. Is the manuscript presented in an intelligible fashion and written in standard English?

Reviewer #1: Yes

Reviewer #2: Yes

5. Review Comments to the Author

Reviewer #1: During your introduction I read about the relationship between gender and food insecurity, nevertheless, this point was not fully addressed during your discussion and how interacts with other related factors.

It is important to include how your results compare to the national ones (as you used the same scale used in the national surveys), from this information it is easier to understand the results with this population.

In general terms, I considered that regarding the information that you had the opportunity to collect you are not fully explaining some interactions between food insecurity and social determinants of health. For example, it is interesting that 55% of your population is above the average schooling level while 68.8% experienced some degree of food insecurity did you check on your models how these two factors interact? Are they related? I encouragingly suggest going deeper inside those SDH that is related to food insecurity then you can explain more about how it is for these group the experience food (in)security.

It is a fact that during your study, there was not a direct correlation between prejudice and food insecurity. Nevertheless, it must be taken into account that food insecurity is a multifactorial problem, and those related to social exclusion are considered part of the causes. I suggest including as a recommendation for further studies to use methodologies that allow showing how prejudice impacts the factors mentioned in line 244 and, as a result, the level of food insecurity in this group.

Reviewer #2: Food insecurity in a Brazilian transgender sample during the COVID-19 pandemic (PONE-D-23-00876)

The manuscript describes results from a cross-sectional, online survey of transgender and gender diverse adults (N = 109) recruited in Brazil. Participants answered questions about food insecurity over the past 3 months, economic factors, demographic information, and perceptions of prejudice. Data were collected between October and December 2020 so participants were reporting about food insecurity relatively early in the COVID-19 pandemic. Results showed high rates of food insecurity -- 68.8% experienced food insecurity, and of these, 20.2% experienced severe food insecurity. Food insecurity was not statistically related to perceptions of prejudice but was predicted by several economic factors including employment status, income, and education. Results were generally expected but the prevalence of food insecurity in this population at this critical time are of value.

Strengths of the manuscript include the investigation of an important problem in a vulnerable population. Limitations include some aspects of the description of the methodology and presentation of findings.

1. In several analyses, such as in Table 1, participants are categorized as having food security or having food insecurity. It was not entirely clear how these were defined. Figure 2 shows that around 30% of the individuals in the food security group reported mild food insecurity and around a similar percentage in the food insecurity group reported mild food insecurity – the latter group also includes individuals with moderate or extreme food insecurity.

2. In the first half of the manuscript, it indicates that “internalized prejudice” was assessed. This normally refers to the attitudes or emotional states of an individual toward themselves which is a reflection of negative attitudes found in the person’s family, community, or country. An example would be a trans person who believes trans people are bad or inferior or has a low self-esteem as a result of being trans because that is what the person has been told by their society.

However, the items in that measure appear to assess the experience of interpersonal discrimination – e.g., from employers, police, or others, rather than the internal state of the individuals.

The second half of the manuscript generally refers to this variable as “perceived prejudice” which seems the more appropriate term. It would be preferable to use just one term for this variable consistently throughout the manuscript.

3. Table 1 shows a marginal association between prejudice and food insecurity (p = .08) and the distributions seem quite different. Given the importance of this variable, were additional analyses conducted, such as assessing if prejudice was associated with extreme food insecurity rather than just food insecurity overall?

4. A minor point, but the Table 1 header does not appear to actually describe the data presented in that table.

6. PLOS authors have the option to publish the peer review history of their article (what does this mean?). If published, this will include your full peer review and any attached files.

Reviewer #1: No

Reviewer #2: No

---

## [Author Response · Author response to Decision Letter 0]

23 Mar 2023

Reviewer's Responses to Questions

Reviewer #1: 

1) During your introduction I read about the relationship between gender and food insecurity, nevertheless, this point was not fully addressed during your discussion and how interacts with other related factors. It is important to include how your results compare to the national ones (as you used the same scale used in the national surveys), from this information it is easier to understand the results with this population.

Answer: Done. We included more information on gender and food insecurity, considering the data at the national level. See the third paragraph of the discussion.

2) In general terms, I considered that regarding the information that you had the opportunity to collect you are not fully explaining some interactions between food insecurity and social determinants of health. For example, it is interesting that 55% of your population is above the average schooling level while 68.8% experienced some degree of food insecurity did you check on your models how these two factors interact? Are they related? I encouragingly suggest going deeper inside those SDH that is related to food insecurity then you can explain more about how it is for these group the experience food (in)security.

Answer: Done. See paragraph 5 of the discussion.

3) It is a fact that during your study, there was not a direct correlation between prejudice and food insecurity. Nevertheless, it must be taken into account that food insecurity is a multifactorial problem, and those related to social exclusion are considered part of the causes. I suggest including as a recommendation for further studies to use methodologies that allow showing how prejudice impacts the factors mentioned in line 244 and, as a result, the level of food insecurity in this group.

Answer: Done. We included these issues in the conclusion.

Reviewer #2: 

4) In several analyses, such as in Table 1, participants are categorized as having food security or having food insecurity. It was not entirely clear how these were defined. 

Answer: Done. We specified this information in the second paragraph of the Instruments and variables, methods section.

5) Figure 2 shows that around 30% of the individuals in the food security group reported mild food insecurity and around a similar percentage in the food insecurity group reported mild food insecurity – the latter group also includes individuals with moderate or extreme food insecurity.

Answer: In figure 2, the first column is about just people who experienced food security, and the second column is about just people who experienced food insecurity, so in the food insecurity group, it is true that around 30% reported mild food insecurity. We perceived that the color was incorrect in the first column and changed it.

6) In the first half of the manuscript, it indicates that “internalized prejudice” was assessed. This normally refers to the attitudes or emotional states of an individual toward themselves which is a reflection of negative attitudes found in the person’s family, community, or country. An example would be a trans person who believes trans people are bad or inferior or has a low self-esteem as a result of being trans because that is what the person has been told by their society. However, the items in that measure appear to assess the experience of interpersonal discrimination – e.g., from employers, police, or others, rather than the internal state of the individuals.

The second half of the manuscript generally refers to this variable as “perceived prejudice” which seems the more appropriate term. It would be preferable to use just one term for this variable consistently throughout the manuscript.

Answer: That true. We adopted perceived prejudice in the entire manuscript.

7) Table 1 shows a marginal association between prejudice and food insecurity (p = .08) and the distributions seem quite different. Given the importance of this variable, were additional analyses conducted, such as assessing if prejudice was associated with extreme food insecurity rather than just food insecurity overall?

Answer: We agree, and we tried to conduct analyses to test the association between prejudice and extreme food insecurity, but because of the number of categories in both variables (perceived prejudice and food insecurity), the number of cases in the table is insufficient to perform rigorous statistical analysis. We included in the conclusion the suggestion that it is necessary to conduct research with a bigger sample to test the differences between categories.

8) A minor point, but the Table 1 header does not appear to actually describe the data presented in that table.

Answer: Done.

---

## [Editor Report · Decision Letter 1]

27 Mar 2023

Food insecurity in a Brazilian transgender sample during the COVID-19 pandemic

PONE-D-23-00876R1

Dear Dr. Savio Marcelino Gomes

We’re pleased to inform you that your manuscript has been judged scientifically suitable for publication and will be formally accepted for publication once it meets all outstanding technical requirements.

Kind regards,

Cesar Infante Xibille, Ph.D

Academic Editor

PLOS ONE

---

## [Editor Report · Acceptance letter]

13 Apr 2023

PONE-D-23-00876R1 

Food insecurity in a Brazilian transgender sample during the COVID-19 pandemic 

Dear Dr. Gomes:

I'm pleased to inform you that your manuscript has been deemed suitable for publication in PLOS ONE. Congratulations! Your manuscript is now with our production department. 

Kind regards, 

on behalf of

Dr. Cesar Infante Xibille 

Academic Editor

PLOS ONE